# Treatment of Landfill Leachate by Short-Rotation Willow Coppice Plantations in a Large-Scale Experiment in Eastern Canada

**DOI:** 10.3390/plants12020372

**Published:** 2023-01-13

**Authors:** Patrick Benoist, Adam Parrott, Xavier Lachapelle-T., Louis-Clément Barbeau, Yves Comeau, Frédéric E. Pitre, Michel Labrecque

**Affiliations:** 1Institut de Recherche en Biologie Végétale, Université de Montréal, Montréal, QC H1X 2B2, Canada; 2Groupe Ramo, 457 Rang du Ruisseau des Anges Sud, Saint-Roch-de-l’Achigan, QC J0K 3H0, Canada; 3Department of Civil, Geological and Mining Engineering, Polytechnique Montréal, Montréal, QC H3C 3A7, Canada

**Keywords:** landfill leachate, short rotation willow coppice, Salix, ammonia and nitrogen removal

## Abstract

The treatment of leachate by vegetative filters composed of short-rotation willow coppice (SRWC) has been shown to be a cost-effective alternative to conventional and costly methods. However, few studies have considered the treatment capability of willow filters at a scale large enough to meet the industrial requirements of private landfill owners in North America. We report here on a field trial (0.5 ha) in which a willow plantation was irrigated with groundwater (D0) or aged leachate at two different loadings (D1 and D2, which was twice that of D1). Additionally, half of the D2-irrigated plots were amended with phosphorus (D2P). The system, which operated for 131 days, was highly efficient, causing the chemical oxygen demand concentration to drop significantly with the total removal of ammonia (seasonal average removal by a concentration of 99–100%). D2P efficacy was higher than that of D2, indicating that P increased the performance of the system. It also increased the willow biomass 2.5-fold compared to water irrigation. Leaf tissue analysis revealed significant differences in the concentrations of total nitrogen, boron, and zinc, according to the treatment applied, suggesting that the absorption capacity of willows was modified with leachate irrigation. These results indicate that the willow plantation can be effective for the treatment of landfill leachate in respect of environmental requirements.

## 1. Introduction

Waste management is a major issue for municipalities and governments around the world [1]. In Canada, more than 25 million tons of municipal solid wastes (MSWs) are generated each year and have to be eliminated. This represents approximately 729 kg cap^−1^ yr^−1^ [2]. One of the major challenges of landfills is the generation of polluted leachate resulting from the percolation of rainwater through the different layers of waste [3]. A multitude of contaminants, including ammonia, dissolved organic matter, phenolic compounds, chlorides, sulfates, trace elements (TE), and xenobiotic organic compounds, can be found in leachate, which can lead to the contamination of ground and surface water near the site [4,5,6,7]. Landfill leachate shows large variability in its pollutant concentration, depending in particular on the age of the waste stored in the landfill cells [8]. In the province of Quebec (Canada), leachate must be captured and treated before it is discharged into the surrounding environment [9]. Conventionally, leachates are treated by expensive physico-chemical, filtration, or biological processes. The current leachate treatment practice in Quebec uses biological processes operated in alternating aerobic or anaerobic modes for carbon, ammonia, and total Kjeldahl nitrogen (TKN) removal [10].

To overcome some of the treatment efficiency and cost challenges encountered in leachate treatment, an innovative approach was proposed and tested on operating landfill sites in Europe over the last twenty years. It consists of leachate treatment systems that combine aerated retention ponds and short-rotation coppice (SRC) using willows or poplars established on the roof of former landfill cells or on adjacent arable land irrigated with leachate [11,12,13,14]. This is a particularly appealing approach. Costs are low since the implementation and treatment both occur on-site [15]. Furthermore, the biomass produced in various sectors of chain values (such as the production of bioenergy or other bioproducts) that are associated with the ecological services offered by such structures (such as CO_2_ capture and storage, visual landscape improvement, and the increase in biodiversity) can be integrated every 3–5 years.

However, to secure these advantages, the phytofiltration systems must be functioning properly, which implies that (i) plant growth is not affected by the nature or quantities of the leachate applied and that (ii) contaminant leaching into the underground water after filtration is reduced or prevented (if on adjacent land). It has been reported that the efficiency of phytofiltration systems was subject to variations depending on plantation design setup and that symptoms of toxicity related to excessive salt concentrations (chloride ions) and nutrient imbalances (nitrogen and phosphorus) have been observed [11,16]. Other factors can also interfere with the effectiveness of leachate treatment, including soil texture, tree species or cultivars, the quality and quantity of the leachate applied, and irrigation methods, which can make it difficult to apply standard operating procedures in various environments [17,18].

Several studies showed that willows could effectively capture nitrogen, phosphorus, organic matter, and certain metals (Cd, Zn, Pb, Ni) in various applications, such as added organic fertilizers [19,20], municipal wastewater [21], or leachate [22]. More recently, Lachapelle-T et al. [23] carried out a pilot project in southern Quebec (Canada) using willows in SRC to treat municipal wastewater and concluded that the system allowed the removal of more than 95% of the nitrogen and phosphorus contained in the effluent. More importantly, the study also demonstrated that such plantations, characterized by very high evapotranspiration rates, could receive up to 25 mL ha^−1^ yr^−1^ of wastewater without negative impacts on willow growth or the environment [24]. Since landfill leachate has a high and variable contaminant concentration, little is known about the treatment ability of willows in SRC. 

The objectives of this study were (1) to determine the efficacy of an SRC willow plantation when used as a vegetative filter in the pedoclimatic conditions of Quebec (Canada) for treating various loads of leachate from inoperative landfill cells and (2) to verify the impacts on soil, runoff water, and willow yields.

## 2. Results

### 2.1. Water Balance

#### 2.1.1. Rainfall, Leachate and Water Irrigation

Rainfall rates measured in 2020 were below the climate normal 1980–2005 (Environment Canada, 2020), particularly from May to early July (Figure 1A). At the same time, the cumulative reference ETo has been estimated at 796 mm: a significant value that could be attributed mainly to the early heat waves observed in June and early July 2020 (Figure 1B). Data calculated from the evapotranspiration model showed D2, D2P > D1 > D0 rates reaching a peak in mid-August (10.7, 9.6, and 5.5 mm/day, respectively; Figure 1C), indicating that leachate irrigated willows evapotranspired about twice as much as water irrigated ones (D0). Leachate and groundwater irrigation was applied for 131 days in 2020 (from 4 June to 12 October). Under our experimental conditions, the average irrigation target rate of 100% (3 m^3^ day^−1^ for D0 and D1; 6 m^3^ day^−1^ for D2) was achieved on 49 days during the season. In total, the four irrigated experimental plots per treatment (804 m^2^) were fed with 237–244 m^3^ (D0–D1) or 439–453 m^3^ (D2–D2P), corresponding, respectively, to 2994 m^3^ ha^−1^ and 5561 m^3^ ha^−1^ (Figure 1D).

#### 2.1.2. Root Zone Depletion Model

Using the RZDM results, the quantity of water remaining in the soil, combining irrigation, rainfall, and evapotranspiration, was determined for each treatment (Figure 2). Soils treated with leachate (D1, D2) had lower moisture levels when compared to those treated with water only (D0). In addition, high volumes of irrigation (D2), combined with high rainfall events that occurred in July and August, decreased water depletion to zero on four occasions during the summer. 

### 2.2. Soil Analytical Results

#### 2.2.1. Ammonia and Nitrate Content

Ammonia availability in the soil remained low during the growing season and did not vary significantly among D1, D2, and D2P treatments despite the high ammonia concentration in the leachate used for irrigation (Table 1). The average concentration of ammonium in the soil ranged from 7.5 to 9.5 mg kg^−1^ dry weight and was similar to the control treatment D0 (8.8 kg^−1^ dry weight) (Figure 3A), but no statistical difference was found between the treatments. The average nitrate concentration in the soil was low in June prior to irrigation (Figure 3B). After four months of irrigation (end of October), a significant increase in nitrate concentration was observed only in the soil treated with leachate. D0 was statistically different from D1, D2, and D2P. The relative stability in ammonia concentration throughout the growing season with the concomitant increase in nitrate was most likely due to aerobic bacterial nitrification and plant uptake.

#### 2.2.2. Micro- and Macronutrient Soil Content Variation

Soil chemical analyses were conducted before (June) and after (October) the irrigation of willows to investigate the effect of each treatment on the variation of mineral concentrations in the soil at the end of the irrigation period. Although data presented in Table 1 revealed a relative variability in soil mineral concentration patterns, some tendencies emerged when each treatment was analyzed individually (June vs. October). The concentration of As, Cr, Mg, and Pb increased in all treatments, while B, Cd, and K concentrations increased only in D1, D2, and D2P treatments. The concentration of Ca decreased in all the treatments except D0. The concentration of the remaining minerals (Al, Ba, Co, Fe, Mn, Ni, P, Se, Zn) showed slight or no variation during the growing season. The total phosphorus concentration was higher in D2P than in D2 in June. A decrease in phosphorus was observed in October with D1 and D2P treatments but not with the D2 treatment. The application of a 2X leachate irrigation volume (D2/D2P) in the soil did not double the concentration of nutrients in the soil. K and B were the only nutrients for which a significant difference was observed between leachate-irrigated and water-irrigated treatments. In October, their concentration was around two to five times higher in leachate-treated willows than in water-treated ones. 

### 2.3. Porewater Analysis

#### 2.3.1. Chemical Oxygen Demand

The average seasonal influent concentration of chemical oxygen demand (COD) was 288 ± 43 mg L^−1^ in leachate and 9.8 ± 6.1 mg L^−1^ in groundwater. COD applied to D1; D2-D2P was 20 or 40 times more than in D0, respectively. The statistical analysis of porewater data revealed significant differences between D1, D2, and D2P treatments. The best removal performance was with D1 (80%), compared to D2 (42%) or D2P (62%), while technically, no removal was observed with DO treatment (Table 2). Interestingly, the percentage of removal for COD remained stable over the season (data not shown).

#### 2.3.2. TKN and Ammonia 

The average seasonal influential concentration of TKN was 0.22 mg/L in groundwater and 221 mg/L in leachate: very similar to that measured for NH_4_-N. This indicated that TKN was mainly added to plots by leachate and that the main nitrogen source present in leachate was NH_4_-N. Although leachate irrigated plots received between 967 (D1) and 1824 (D2 and D2P) times more TKN than groundwater irrigated ones, the efficiency of NH_4_-N removal was high for D1, D2, and D2P treatments: nearly 100%, approaching the NH_4_-N concentration measured in the porewater of D0 treatment (Table 2). For each treatment, the percentage of NH_4_-N removal remained stable over the growing season (data not shown).

#### 2.3.3. Variation in Micro- and Macro-Element Porewater Content

A reduced number of minerals were analyzed in the porewater collected at two irrigation times (mid-season in August and end of the season in October). The total phosphorus concentration decreased throughout the season. However, the addition of phosphorus fertilizer in treatment D2P did not appear to significantly increase the porewater phosphorus content (Table 3). As a general rule, the concentration of Ca, Fe, Mg, Na, and K increased progressively in the porewater during the growing season for D2 and D2P treatments. Conversely, the concentration of these elements decreased in the D1 treatment except for Ca. Interestingly, the concentration of Fe was lower in leachate treatments than in the water treatment. Except for P, the data clearly distinguished two statistical groups (D1/D2/D2P and D0). Other elements (Al, Cu, Fe, Mn, Ni, and Zn) had no clear trend, and treatments could be discriminated against (data not shown). 

### 2.4. Plant Analysis

#### 2.4.1. Biomass Productivity

Very few willows died after the first growing season following their establishment in 2018. After two years of growth (2020), the average final yields of the aboveground biomass were 44.0, 32.8, 18.2, and 17.1 tons of dry mass per hectare (tons dry weight/ha) for D2P, D2, D1, and D0, respectively (Figure 4). Leachate irrigation had a significant effect on the biomass yield (*p* = 0.0136), which was, on average, 2.5-fold higher in D2P in comparison to D0. 

#### 2.4.2. Leaf Surface Area

The leaf surface area, from the largest to smallest area, ranked as follows: D2, D2P > D1, D0 (Figure 5A). Statistical analyses revealed that willows irrigated with the highest leachate volumes (D2, D2P) had larger leaves. However, phosphorus fertilization did not significantly affect the leaf surface area since similar yields were observed for D2 and D2P. 

#### 2.4.3. Leaf Pigment Content

Greater variability in the chlorophyll (a + b) and carotenoid concentrations were observed in the leaves of willows irrigated with water (Figure 5B,C). Pigment concentration increased during the summer, peaking in August for all treatments before declining notably in October. Peaks were significantly higher for leachate-irrigated D1, D2, and D2P treatments.

#### 2.4.4. Leaf Nutrient Analysis

A chemical analysis of the leaves and stems harvested at the end of August 2020 was carried out in order to determine the concentration of a limited range of nutrients. A higher nitrogen percentage (between 56 and 86%) was found in the leaves from leachate-irrigated willows than in the leaves of plants irrigated with water (Table 4). A similar finding was observed for B, distinguishing two statistical groups, D0/D1 and D2/D2P, with an increase of about 40% for B in the second group. On the contrary, Zn was at a lower concentration in the leaves of leachate-irrigated willows (between 53% and 62% for the D1/D2/D2P statistical group). There was no clear statistical difference between treatments for the other elements. 

## 3. Discussion

Many studies have demonstrated that willow vegetation filters can effectively treat various sources of wastewater on-site under a diversity of climates [25,26,27,28,29]. In this study, up to 7.5 mm d^−1^ (D2/D2P treatments) of leachate was applied to willows (*Salix miyabeana* ‘SX64’) grown on the clay layer topping an inoperative landfill cell. Over a cycle of 131 days, from June to October, plants were irrigated for about 110 days, thereby showing that willows are well adapted to water-saturated and compacted soils, with no negative impact on their growth and development [12,30,31]. Similar large-scale studies (thousands of m^2^) were carried-out with *Salix miyabeana* ‘SX67’ under Quebec’s humid continental climate to treat primary municipal wastewater [20,22]. Cultivar ‘SX67’, a cultivar of the same willow species utilized in this study, has been shown to have a very high-water demand, allowing a high irrigation rate of up to 2961 mm yr^−1^ with a former agricultural soil, which enables the treatment of up to 30,000 m^3^ ha^−1^ yr^−1^ of municipal wastewater [23,32]. However, it was shown that high hydraulic loading rates (HLR) were applied to willows at the end of the season, while evapotranspiration was low and growth started reducing, causing deep percolation and nitrogen leaching into the groundwater, indicating that HLR should be modulated according to its willow filtering capability [33].

Evapotranspiration is one of the most critical factors in large-scale industrial plant-based treatments whose main objective is to minimize the wastewater volumes to be treated conventionally [34]. Willows can be classified among the species displaying high evapotranspiration rates [35]. Under the climate of Eastern Canada, many willow cultivars remain photosynthetically active very late in the season, when the majority of deciduous species have lost their foliage [36]. This is another advantage of using willows in the treatment of wastewater or leachate. In the current study, the vegetation filter was able to treat, by evapotranspiration, up to 5560 m^3^ ha^−1^ of leachate, exhibiting moderate electroconductivity (3.7–4.2 mS cm^−1^). However, the RZDM model predicted that, over the growing season, an average of 4.5% of the total water (rainfall and irrigation) received by willows would potentially end up as runoff outside the operational filtration unit (data not shown). This water balance model also revealed root zone depletion during most of the irrigation campaign (between 10 and 70 mm for D1 and D2/D2P; also validated by in situ soil moisture tension results; data not shown), indicating that additional work would be needed to maximize the retention capability of the clay soil used in the cultivation of such landfill leachate filter systems. The relative similarity in the RZDM pattern between D1 and D2 treatments, in contrast to the D0 treatment, could be partially explained by the gain in the aboveground biomass production of leachate-irrigated willows, which promotes the evapotranspiration rate of plants as described above and, consequently, decreases soil moisture.

A significant increase in the biomass yield following leachate addition is reported here. The average biomass production in our experimental conditions after two years reached 18 tons ha^−1^ for intermediate (D1) loading but exceeded values of 44 tons ha^−1^ for the D2P treatment. In contrast, the biomass yield was only 17 tons ha^−1^ in the groundwater treatment (D0). However, these values remain less impressive than those obtained by Jerbi et al. [32]. In that study, carried out in a similar climatic context but on agricultural land, annual yields of 40 tons ha^−1^ y^−1^ were obtained when the willows were irrigated with wastewater from municipal primary effluents. It is likely that the soil conditions, as well as the various characteristics of the two effluents, explain this difference. Still, leachate had a positive effect on willow growth and induced no signs of toxicity. Apart from the water supply, fertilization is described as a driving factor that influences biomass yield and, consequently, evapotranspiration [29]. Jerbi et al. [32] have shown that high nitrogen inputs to the soil by wastewater irrigation resulted in a significant increase in stomatal conductance, leaf area, as well as chlorophyll and nitrogen leaf content, while the fine root:aboveground biomass ratio decreased [32,37]. Our data closely match these findings since leaf area and pigment content was significantly higher with the highest leachate supply (providing 1 478 kg ha^−1^ NH_4_-N cumulative loading over the growing period) compared to water treatment. Moreover, the leachate-irrigated willows maintained their green pigments and leaves for approximately two to three weeks longer than the control treatment at the end of the growing season, which may be part of the reason for their additional growth. It would also explain why the maximal evapotranspiration rate measured in August was two times higher in D2-treated plants (10.7 mm day^−1^) than in D0-treated ones (5.5 mm day^−1^). Finally, the high nitrogen supply to the soil during leachate irrigation significantly increased the growth of the longest willow stems by the end of the growing season, contributing to a higher aboveground biomass yield. 

In the current study, different removal patterns were observed, with leachate contributing up to 1907 kg COD ha^−1^ yr^−1^ and 1478 kg NH_4_-N ha^−1^ yr^−1^. Removal varied, on average, from 42% (D2) or 62% (D2P) to 80% (D1) for COD and was close to 100% for NH_4_-N. Furthermore, the system’s filtration capacity remained effective throughout the irrigation period. The high efficiency of willow vegetation filters for removing pollutants from wastewater is well documented. A trial conducted in Sweden in 2010 showed a removal efficiency of about 80% for organic carbon (TOC) and total N by the cultivar ‘Tora’ (*S. schwerinii x S. viminalis*) irrigated with leachate whose total loadings applied were 378 kg ha^−1^ yr^−1^ and 2021 kg ha^−1^ yr^−1^ for the total N and TOC, respectively [22]. More recently, Jerbi et al. [32] reported high removal efficiencies (93.4% for COD, 99.3% for NH_4_-N, and 98.3% for P) with *S. miyabeana* ‘SX67’ irrigated with municipal wastewater contributing to 8700 kg COD ha^−1^ yr^−1^, 1245 kg N ha^−1^ yr^−1^ and 121 kg P ha^−1^ yr^−1^. While willows can take up NH_4_-N for nutritional purposes, a bacterial nitrification process could also explain its removal efficiency. The increase in NO_3_–N in the superficial soil horizon was between 3.75-fold for D1 and 6.0-fold for D2/D2P at the end of the growing season. It cannot be excluded that the nitrification process could be underestimated due to the uptake by willows of this preferred nutritional nitrogen form from the soil [38]. Interestingly, we have noticed that the NO_3_–N concentration decreased by about 2–3 fold in May 2021 in leachate irrigated soil (data not shown), suggesting that a nitrification process could occur during the winter (from November 2020 to May 2021) before initiating a new irrigation cycle. Jones et al. [18] pointed out that highly concentrated leachates (EC > 4.0 mS cm^−1^) could affect the filtering capacity of the plant system, thereby limiting the volumes that could be processed (<250 m^3^ ha^−1^ yr^−1^). This could be one of the explanations for the lower COD removal observed in the porewater of the D2 leachate treatment (EC 4.2 mS cm^−1^). However, the filtering capacity of our willow filtering system was not adversely impacted by this EC range for the removal of NH_4_-N, regardless of the leachate dose applied to the willows. This finding concurs with data obtained with the leachate-irrigated *S. matsudana* ‘Levante’, indicating that the NH_4_-N concentration significantly decreased under the lowest leachate supply (i.e., around EC 4.0 mS/cm; 29). Moreover, concentrations detected in the soil porewater after irrigation with moderately concentrated leachate (i.e., 111 mg COD L^−1^ and 0.82 mg NH_4_-N L^−1^ for D2P treatment) did not exceed the standard limits imposed by Quebec regulations, which are 25 mg NH_3_-N L^−1^ and 150 mg BOD_5_ L^−1^ for the release of treated leachate in the environment [39].

Phosphorus fertilization improved leachate treatment efficiency, increasing COD, TKN, and NH_4_-N removal by 44%, 32%, and 64%, respectively. Soil matrix physico-chemical characteristics of the site (i.e., high clay content, low organic matter percentage, low initial phosphorus soil concentration, and soil compaction) could negatively affect the phosphorus availability of the soil and, therefore, its assimilation by plants. Furthermore, leachate added very low amounts of phosphorus to the irrigated willow filters. Consequently, one treatment (D2P) was added to the experimental design, consisting of a phosphorus amendment of 40 kg P_2_O_5_ ha^−1^ yr^−1^ when applied to willows irrigated with the highest leachate dose (D2 treatment), which corresponds to willows requirements in SRC [40,41,42]. Phosphorus is essential to growth and root proliferation at the initial stages of establishment. When associated with nitrogen, it is highly effective on willow aboveground and with a root-dry biomass yield [43]. In the present study, higher aboveground biomass yield (+25%) and leachate treatment efficiency (+44% for COD, +32% for TKN, and +64% for NH_4_-N compared to D2 treatment) were achieved with P fertilization. These findings suggest that phosphorus could influence the soil bacterial community, especially species involved in degrading the organic matter and nitrogen present in the leachate. Indeed, previous research has reported phosphorus availability to be one of the major variables regulating the abundance and diversity of the bacterial community, and this shaping occurred only when phosphorus was associated with carbon or nitrogen [44,45,46].

In the present study, higher amounts of nutrients were measured in leaf tissues after leachate irrigation (especially B, Fe, and N). The significant strong increase in the N content in leaf tissues of leachate-irrigated willows (+56% in D1, +86% in D2, and +76% in D2P) was likely correlated with the high amount that nitrogen leachate contributed to the soil, thus suggesting the better availability and recovery of essential nutrients by plants for their growth. The high removal efficiency of nitrogen achieved by willows has also been documented in other studies [21,22,32,47]. Nutrient interactions in crop plants are multiple and complex, acting at several physiological levels, including transportation and signaling [48]. They may induce deficiencies or toxicities, modify growth responses, and/or modify nutrient composition [49]. For instance, a positive interaction between phosphorus and nitrogen was reported in the context of their mutual absorption by plants and synergetic effects on plant stimulation. On the other hand, applied high levels of phosphorus may induce an iron deficiency in plants [49]. In the context of the phytoremediation described here, a well-balanced supply of nutrients could be critical for optimizing both leachate treatment and the production of renewable high-quality aboveground wood biomass (willows are usually coppiced every three years) for commercial use. 

Although the leachates used in this pilot study exhibited moderate sodium and chloride concentrations (around 300 mg L^−1^) and EC, we observed that the EC of porewater increased gradually over the course of the season with leachate treatments, while it remained low and stable with the water control. This increase was faster for D2/D2P than for D1 at the beginning of the irrigation campaign, reflecting the different volumes of leachate applied for irrigation. At the end of the irrigation period, it was 3.1, 4.2, and 3.7 mS cm^−1^ for D1, D2, and D2P, respectively, thus closely approaching the value measured in the leachate (data not shown). In addition, in the porewater of the D2 treatment, the total sodium reached 402 mg L^−1^ at the end of the irrigation campaign. As such, there is no evidence that this salinity rate was a factor limiting the growth of willows and their efficiency in treating leachate. Nevertheless, from a long-term perspective, the progressive salinization of the soil in which the willows grow could have an antagonist effect on phytoremediation performance. Although several willow cultivars have shown good tolerance to moderate salinity treatment (7 dS m^−1^), Canadian indigenous willows were reported to have a higher potential for long-term survival under severe salinity treatment (14 dS m^−1^) [50]. Moreover, recent works have reported an increased adaptation to soil salinity among tetraploid hybrids, thanks to a different Na^+^ balance between the leaves and roots, which is also characteristic of the cultivar ‘SX64’ used in this study, compared to diploid ones [51].

## 4. Materials and Methods

### 4.1. Site Description

The experiment was conducted at the engineered landfill site operated by waste management (WM) located in the municipality of Sainte-Sophie in Southern Quebec (45°47′02.98″ N, 73°54′12.38″ W) (Figure 6). In early July 2018, a 0.5 ha willow plantation was planted on top of a former (inoperative) landfill cell previously capped with a 1 m high clay layer and grassed in the early 1980s. One-year-old, 20 cm long, unrooted dormant cuttings of *Salix miyabeana* ‘SX64‘ were planted at a density of 16,000 plants/ha (in double rows spaced 1.5 m apart) in early July and coppiced in mid-November 2018. Willow crops were irrigated with leachate from the former landfill cells or groundwater for the two subsequent growing seasons (2019–2020). The climate of Sainte-Sophie is characterized by humid cold winters and hot, humid summers with a monthly average temperature of −12.7 ± 3.1 °C (January, coldest month) to 19.5 ± 0.9 °C (July, hottest month) over a 20-year period [52]. Weather conditions, including wind speed, humidity, rainfall, temperature, pressure, and solar radiation, were measured at 5 s intervals by a Vantage Pro2 Plus weather station, which was installed on site according to the manufacturer’s instructions (Davis Instruments Corp., 2004). Rain on site for the May-October growing seasons in 2019 (479 mm) and 2020 (526 mm) was below the climate normal for the same period (556 mm). 

### 4.2. Experimental Setup and Willow Treatment

The 0.4 ha strip-block experimental setup was divided into four blocks (B1–B4) comprising a total of 20 equally sized 9.14 m × 22 m rectangular plots, with four treatments (D0, D1, D2, and D2P) replicated four times (Figure 1). Each plot was assigned one of four treatments corresponding to a maximum irrigation rate of 3 m^3^ day^−1^ of leachate for D1, 6 m^3^ day^−1^ of leachate for D2, 6 m^3^ day^−1^ for leachate, plus 40 kg P_2_O_5_ ha^−1^ yr^−1^ of phosphorus fertilizer (D2P) or 3 m^3^ day^−1^ of water for D0. Plot assignments for treatments D1 and D0 were selected based on the irrigation lines system design, while plot assignments between treatments D2 and D2P were randomized (Figure 6). The experimental area was buffered by outer rows of willows of the same width to prevent edge effects and was surrounded by an unplanted zone 8–10 m wide containing a polymer-lined soil berm to collect rainwater and snowmelt runoff. Leachate was pumped from former (inoperative) landfill cells and diverted into an RT1 storage reservoir (5800 L). The stored leachate was then pumped to the irrigation staging area, which was equipped with a decanting tank and a leachate storage tank (5800 L). Water was pumped from groundwater wells to another storage tank (5800 L) at the irrigation staging area (Figure 7). The irrigation system consisted of four distribution lines (one for D0 and D1; two for D2) which were installed with irrigation sprinklers every 7 m. An irrigation control and recording system comprising flowmeters, pressure gauges, and programmable valves were installed to modulate the volume of leachate/water to be applied to the experimental area according to rainfall levels and the evapotranspiration capacity of willows during the growing season. One suction-cup lysimeter (model 1900, Soil Moisture Corp.) was installed at a depth of 50 cm in the middle of each plot to collect soil porewater. Two irrigation campaigns were conducted during the growing season from 2019 to 2020 (88 and 131 days of irrigation, respectively), but only data for the second year are presented.

### 4.3. Soil, Leachate, Groundwater and Porewater Sampling and Analysis

An initial characterization of the soil surface (0–30 cm) was performed prior to irrigation in early July 2019, consisting of four samples per block, each of which comprised 12 homogenized subsamples: four per plot (Table 5). In 2020, the surface soil was sampled twice (before and after the growing season, i.e., early June and late October). Soil samples were collected using a soil core sampler, previously washed with alconox detergent, and rinsed with distilled water between each plot sampling. Pre-irrigation and post-irrigation sampling consisted of 16 analytical samples (one per plot), each comprising four homogenized subsamples collected from each plot (n = 4). For some analyses, sampling consisted of four analytical samples (one per treatment), each comprising three homogenized subsamples collected from each plot (n = 1). These samples were analyzed for pH, macronutrients (NH_4_-N, NO_3_-N, Ca, K, Mg, Mn, P), and trace elements (Al, As, B, Ba, Cd, Co, Cr, Cu, Fe, Pb, Se, Zn). Phosphorus, K, Ca, Mg, Al, Mn, Cu, Zn, B, and Fe were extracted by the Mehlich III method. A total acid extraction method was used for the other trace elements.

The leachate and groundwater used for irrigation and the soil porewater were sampled every 2–3 weeks from the start of irrigation to the end of the growing season for 10 campaigns. Leachate and groundwater were sampled directly from the storage tanks for total suspended solids (TSS), volatile suspended solids (VSS), biochemical oxygen demand (BOD_5_), chemical oxygen demand (COD), ammonia nitrogen (NH_4_-N), nitrate nitrogen (NO_3_-N), total Kjeldahl nitrogen (TKN), total and orthophosphates, Ca, Cl, Fe, K, Mg, Na, Zn, total phenolic compounds, pH, electroconductivity (EC), oxidation-reduction potential (ORP), and total dissolved solids (TDS). pH, ORP, EC, and TDS were measured in the field using a Hanna combo tester (model HA-98129 from ERE Inc. Saint-Leonard, QC, Canada). The rest of the parameters were analyzed in a certified laboratory. The main physicochemical characteristics are presented in Table 6.

Soil porewater was sampled from lysimeters, which were depressurized to −65 centibars (cb) between 24 and 48 h in advance of the sampling by extraction with a low-flow peristaltic pump. The samples were analyzed for COD, NH_4_-N, and NO_3_-N in eight campaigns and for alkalinity, both dissolved and in total metals (Ca, Fe, Mg, K, Na, and Zn).

### 4.4. Plant Biomass

Biomass productivity was measured at the end of the season by weighing three randomly selected fresh trees per plot and per treatment (48 trees in total, cut at 10 cm from the ground surface) which were weighed on-site to determine the wet biomass using a portable electronic scale (GRAM SAFIR X50, Barcelona, Spain). Dry biomass was calculated based on a dry/wet ratio determined by drying wood samples at 70 °C for at least 72 h.

### 4.5. Plant Sampling and Analysis

The longest stem length of six randomly preselected willows (A-F) per experimental plot and per treatment was measured twice in 2020 (June and October). Three of these preselected willows (A-C) were used for plant analysis. For chlorophyll content determination, ten leaves were collected from each plant three times (mid-July, mid-August, and early October), starting at the fourth leaf from the tip of the longest stem, which was pooled and maintained in a cooler and then stored in a freezer at −80 °C until analysis using the method of Hiscox and Israelstan [54]. Twenty-five leaves of each plant were harvested in mid-July, five of them for measuring the leaf area index (WinFOLIA software program) and the rest determining the N, P, K, Mg, S, Al, B, Cu, Zn, Fe, Mn, and Zn leaf content based on the Mehlich extraction method. Before nutrient determination, the leaves were pooled by treatment and by block and kept in a cooler.

### 4.6. Data Analysis

A hydrological balance model was developed for the willow plantation based on a daily root zone depletion model (RZDM) [55]. The model consisted of a unit volume of soil with the inputs and outputs calculated for each day the system was in operation, readily available for the water (RAW), and total available water (TAW). The capillary rise was assumed to be zero. Irrigated volumes of water or leachate were measured by the electronic flowmeters installed downstream of the pumps. Rainfall, solar radiation, temperature, and wind speed were measured by the on-site weather station. Runoff volumes were calculated based on the soil conservation service (SCS) method for clayey soils [56]. Deep percolation was the unknown variable calculated in the water balance. Initial depletion was assumed to be zero (saturated condition) based on snowmelt, cool temperatures, and rainfall in May.

The reference evapotranspiration (ETo) was estimated by the Penman-Monteith equation [57], and the evapotranspiration estimate was calculated by multiplying the reference evapotranspiration by the monthly average crop coefficient for willows. The unfertilized treatment (D0) used the unfertilized willow crop coefficient, while the fertilized treatments (D1, D2, D2P) used the fertilized crop coefficient from Guidi et al. [58].

The ability of the vegetative filter to degrade or remove pollutants (i.e., treatment efficacy by concentration) was determined by comparing their concentration in the last leachate (for D1, D2, and D2P) or groundwater (D0) to the ones in the soil porewater using a relative percent difference (% RPD), calculated after each sampling. The last leachate referred to the one applied to willows just before soil porewater sampling. The treatment efficacy for the whole season was estimated by averaging the % RPD of each treatment. 

### 4.7. Statistical Analysis

Groundwater, soil, and other experimental data were analyzed by a two-way analysis of variance, with no interaction statistical model, by using JMP (version 14). The models had concentration as the dependent variable and treatment and block as the independent variables. Biomass productivity data were analyzed by a two-way analysis of variance, with an interaction statistical model, and also by using JMP 14. The model had biomass productivity as a dependent variable and treatment, block, and treatment*block as independent variables, of which block and treatment*block were considered to have a random effect. In all cases, Tukey’s honest significant difference (HSD) post hoc tests were performed to determine the significance of treatment effects at the traditional threshold of 5%.

## 5. Conclusions

Short-rotation willow coppice filters are a valuable alternative to other current approaches for treating inoperative landfill leachates under a cold humid climate. Under these experimental climatic conditions, willow cultivars (*Salix* ssp.) have a relatively long growing season, making the process potentially economically valuable at an industrial scale and, thereby, more attractive. This represents a less energy-consuming and more sustainable alternative for companies that manage landfill sites. 

The maximum leachate loading for this pre-industrial pilot reached 5560 m^3^/ha in 2020, and we showed that willows were effective at decreasing the concentration of NH_4_-N in the effluent and in the soil. However, the efficacy of the vegetative filter for treating COD tended towards saturation when a 2x leachate loading was used for irrigation. The application of leachate on willow plantations modified the concentration of several nutrients present in the leaves, suggesting altered assimilation patterns. As the willows were established on the roof of a landfill cell, mainly composed of clay, some additional tests should be carried out to improve the texture and nutritional properties of this topsoil, increase the leachate treatment capability of this type of vegetative filtering process, and prevent any off-site leakage. Similarly, the presence of contaminants of emerging concern is not investigated in the current study, as well as the possible land quality degradation by salts and trace elements supplied; these are other factors to be considered in order to ensure the long-term survival of willow shrubs and the functionality of the filtering system when operating at an industrial scale. 

## Figures and Tables

**Figure 1 plants-12-00372-f001:**
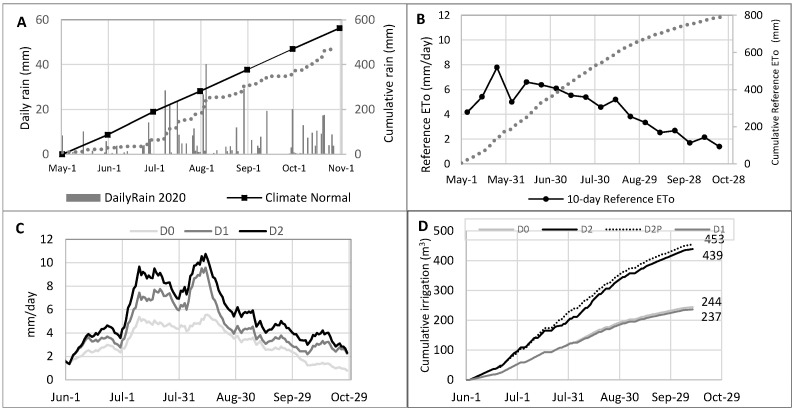
Irrigation parameters in 2020. (**A**) Daily rainfall quantities, cumulative rainfall, and 1980–2005 climate normal; (**B**) A 10-day moving average reference evapotranspiration (ETo) and cumulative ETo; (**C**) Modeled 10-day moving average of evapotranspiration; (**D**) Cumulative daily irrigation volumes (m^3^) applied by treatment (corresponding to an irrigation area of 804 m^2^).

**Figure 2 plants-12-00372-f002:**
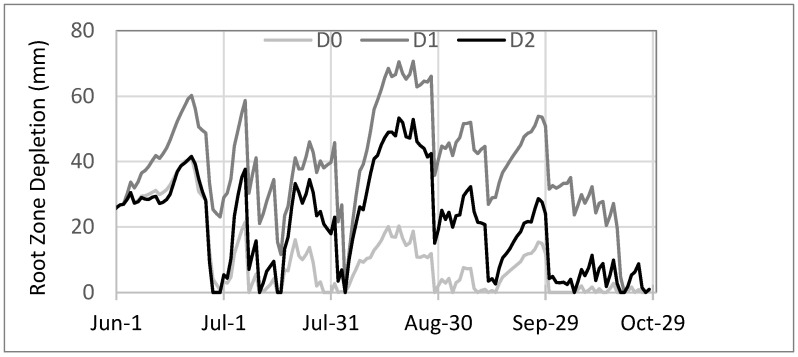
Root zone depletion model applied to irrigated treatments with water (D0) or leachate (D1, D2) in 2020.

**Figure 3 plants-12-00372-f003:**
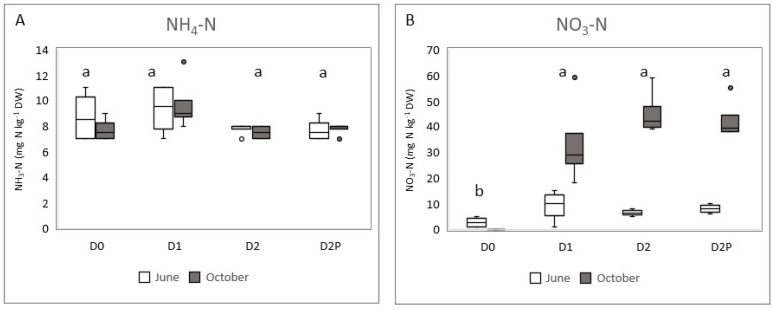
(**A**,**B**): Ammonia (NH_4_-H) and nitrate (NO_3_-N) concentration (expressed in mg N kg^−1^ of dry weight, DW) in the soil before (June) and after irrigation (October). Different letters above groupings indicate significant differences at *p* < 0.05.

**Figure 4 plants-12-00372-f004:**
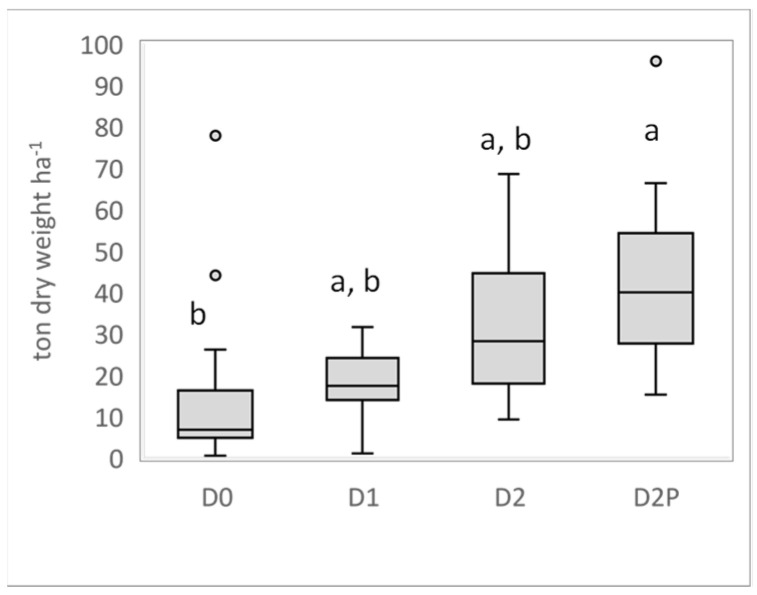
Comparison of the yield measured at the end of the growing season in 2020. Different letters above box plots indicate significant differences at *p* < 0.05. The horizontal line inside each box-whisker represents the median, vertical small rectangles represent interquartile ranges, and circles are data-outliers.

**Figure 5 plants-12-00372-f005:**
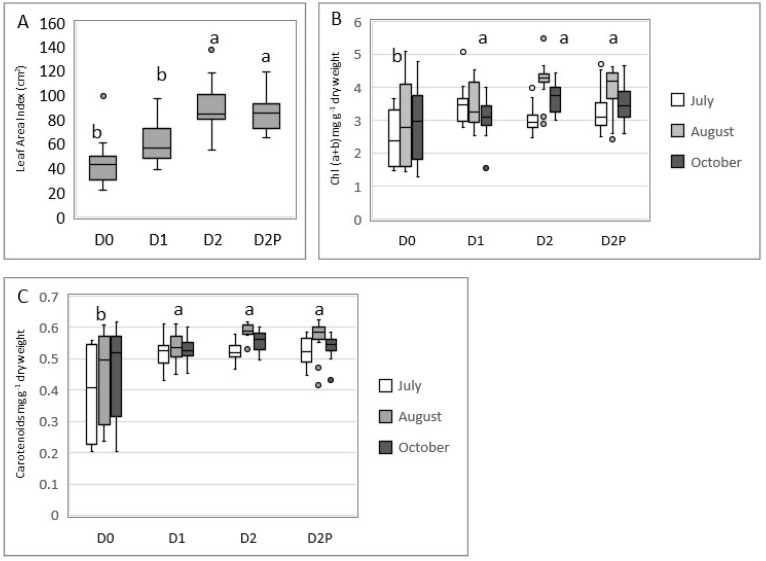
Leaf parameters during the growing season in 2020. (**A**) Average leaf surface area expressed in cm^2^. (**B**,**C**) Average leaf chlorophyll and carotenoids concentrations (expressed in mg g^−1^ dry weight). Lowercase letters indicate statistical groupings (*p* < 0.05). The horizontal line inside each box-whisker represents the median, vertical small rectangles represent interquartile ranges, and circles are data-outliers.

**Figure 6 plants-12-00372-f006:**
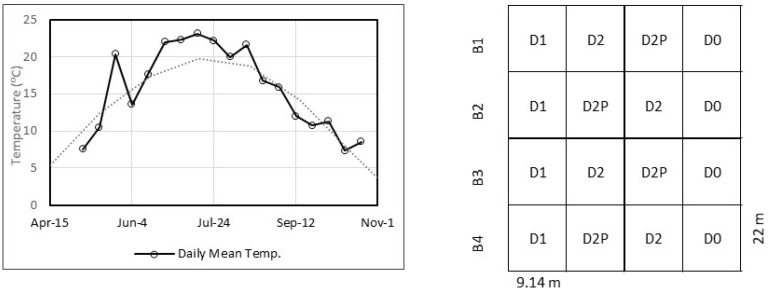
Daily temperature 10-day average for the 2020 season and climate norm 1981–2010, and schematic diagram of the experimental setup.

**Figure 7 plants-12-00372-f007:**
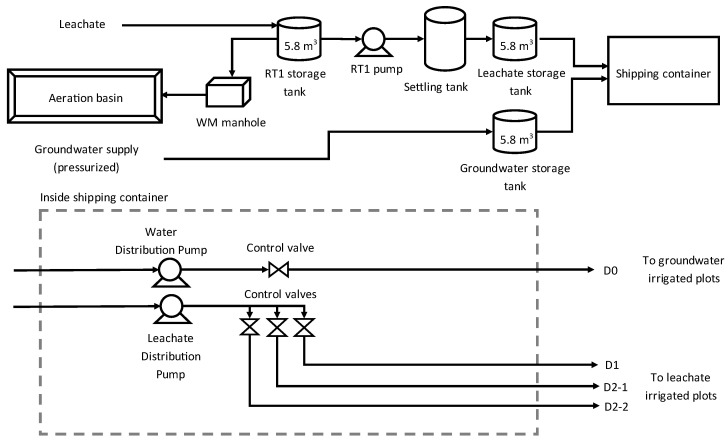
Diagram representing the irrigation system.

**Table 1 plants-12-00372-t001:** Variation in the seasonal concentration of trace elements and macronutrients in the soil before (June) and after (October) irrigation with groundwater (D0) and leachate (D1, D2, and D2P) in 2020.

		Soil Reference		D0		D1		D2		D2P	
Parameters	Units	July 2019	n	June	October	June	October	June	October	June	October
Al	mg/kg	1181 ± 133	4	1255 ± 210	1273 ± 176	1094 ± 141	1024 ± 160	1239 ± 109	1298 ± 207	1304 ± 252	1228 ± 235
As	mg/kg	<5.0	1	0.71	2.21	0.87	1.43	0.99	1.56	1.01	1.94
B	mg/kg	0.5 ± 0.1	4	0.6 ± 0.2	0.4 ± 0.1 ^a^	1.5 ± 0.8	2.0 ± 0.9 ^b^	1.3 ± 0.2	2.9 ± 0.5 ^b^	1.6 ± 0.4	2.3 ± 0.5 ^b^
Ba	mg/kg	198 ± 24	1	140.2	160	162	155.4	132.2	133.6	147.8	146.4
Ca	kg/ha	5793 ± 729	4	5064 ± 1013	5197 ± 963	4508 ± 469	4484 ± 337	4393 ± 1383	4280 ± 1488	4978 ± 879	4700 ± 444
Cd	mg/kg	< 0.50	1	0.0025	0.155	0.57	0.03	0.0025	0.09	0.57	2.27
Co	mg/kg	16 ± 2	1	10.8	12.5	14	13.5	10.2	11.1	9.4	10.7
Cr	mg/kg	81 ± 10	1	56	72	65	71.4	41.5	61.3	49.9	62.4
Cu	mg/kg	10.3 ± 2.6	4	10.8 ± 3.2	10.8 ± 2.3	8.5 ± 1.8	9.1 ± 1.2	8.7 ± 3.7	8.5 ± 4.7	10.4 ± 3.3	10.4 ± 2.0
Fe	mg/kg	345 ± 17	4	340 ± 28	358 ± 30	351 ± 30	372 ± 31	367 ± 63	325 ± 53	328 ± 29	371 ± 26
K	kg/ha	531 ± 104	4	483 ± 88	487 ± 122 ^a^	590 ± 217	898 ± 96 ^b^	522 ± 197	976 ± 308 ^b^	640 ± 133	1002 ± 168 ^b^
Mg	kg/ha	1292 ± 194	4	1041 ± 137	1139 ± 188	1155 ± 217	1362 ± 86	1050 ± 312	1083 ± 304	1097 ± 104	1248 ± 115
Mn	mg/kg	22.1 ± 6.2	4	20.0 ± 11.7	17.0 ± 2.0	24.8 ± 3.2	20.5 ± 4.1	20.9 ± 11.7	16.0 ± 7.2	19.2 ± 5.3	19.5 ± 4.9
Ni	mg/kg	47 ± 6	1	35.9	40.2	43.1	40.4	38	34	26.8	34.2
P	kg/ha	312 ± 208	4	441 ± 239	412 ± 183	141 ± 87	86 ± 71	326 ± 170	341 ± 238	418 ± 291	330 ± 214
Pb	mg/kg	7.0 ± 0.5	1	0.4	8.43	0.05	6.58	0.025	7.44	0.92	9.2

Note: underlined numbers are significantly different. Different letters in rows indicate significant differences (*p* < 0.05); ± indicates standard deviation.

**Table 2 plants-12-00372-t002:** Efficacy of COD and nitrogen removal by the willow filter. Efficacy was calculated as the seasonal concentration average % RPD between influent (groundwater or leachate) and effluent (porewater).

Parameters	Treatment	D0	D1	D2	D2P
COD	Influent (mg L^−1^)	9.8 ± 6.1	288 ± 43	288 ± 43	288 ± 43
	Effluent (mg L^−1^)	28.0 ± 18 *^a^*	58 ± 20 *^b^*	198 ± 69 *^d^*	111 ± 42 *^c^*
	Efficiency %	−185.0	79.8	31.2	61.6
NTK	Influent (mg L^−1^)	0.22 ± 0.06	221 ± 37	221 ± 37	221 ± 37
	Effluent (mg L^−1^)	0.90 ± 0.68 *^a^*	2.3 ± 1.6 *^b^*	5.6 ± 4.0 *^c^*	3.8 ± 2.0 *^c^*
	Efficiency %	−309.1	99.9	97.5	98.3
NH_4_-N	Influent (mg L^−1^)	0.23 ± 0.05	218 ± 35	218 ± 35	218 ± 35
	Effluent (mg L^−1^)	0.06 ± 0.04 *^a^*	0.07 ± 0.08 *^a,b^*	2.29 ± 3.53 *^b,c^*	0.82 ± 1.30 *^c^*
	Efficiency %	74	100	99	99.6

Note: Different letters in rows indicate significant differences (*p* < 0.05); ± indicates standard deviation.

**Table 3 plants-12-00372-t003:** Variation in the seasonal concentration of macronutrients in the porewater of plots irrigated with groundwater (D0) and leachate (D1, D2, and D2P) in 2020.

Parameter	Units	Influent		Effluent							
		Water	Leachate	D0		D1		D2		D2P	
				August	October	August	October	August	October	August	October
**Ca**	mg/kg	78.7	232	139.3 ± 28	151 ± 39 ^a^	344.0 ± 77	368 ± 101 ^b^	396 ± 80	386 ± 108 ^b^	433.0 ± 111	456 ± 116 ^b^
**Fe**	mg/kg	1.45	34.5	3.8 ± 4.3	2.7 ± 1.8 ^a^	0.09 ± 0.07	0.06 ± 0.01 ^b^	0.10 ± 0.09	0.15 ± 0.14 ^b^	0.04 ± 0.02	0.04 ± 0.02 ^b^
**K**	mg/kg	4.5	267.7	8.4 ± 5.7	10 ± 1.6 ^a^	38 ± 22	35 ± 11 ^a^	87 ± 42	119 ± 89 ^b^	49 ± 19	55 ± 36 ^a^
**Mg**	mg/kg	39.2	122.4	62 ± 19	63 ± 22 ^a^	194 ± 63	190 ± 66 ^b^	210 ± 15	201 ± 27 ^b^	212 ± 82	220 ± 73 ^b^
**Na**	mg/kg	21.7	333	12 ± 3	11 ± 5 ^a^	262 ± 180	250 ± 99 ^b^	396 ± 129	467 ± 172 ^b^	371 ± 209	377 ± 210 ^b^
**P**	mg/kg	0.0008	0.006	0.17 ± 0.11	0.09 ± 0.02	0.50 ± 0.55	0.07 ± 0.05	0.14 ± 0.05	0.07 ± 0.04	0.24 ± 0.15	0.13 ± 0.09

Note: underlined numbers are significantly different. Different letters in rows indicate significant differences (*p* < 0.05); ± indicates standard deviation.

**Table 4 plants-12-00372-t004:** Leaf micro- and macro-element concentrations in 2020 expressed as % of dry weight or in mg/kg dry weight with comparison to D0.

Parameters	Units	Treatment						
		DO	D1	% D0	D2	% D0	D2P	% D0
N	%	2.08 *^a^*	3.25 *^b^*	56	3.87 *^b^*	86	3.66 *^b^*	76
P	%	0.26	0.23	−12	0.3	15	0.29	12
K	%	1.53	1.59	4	1.64	7.2	1.66	9
Ca	%	1.43	1.32	−8	1.25	−13	1.23	−14
Mg	%	0.29	0.33	14	0.32	10	0.29	0
S	%	0.36	0.43	19	0.39	8	0.36	0
Al	mg/kg	22.5	27	20	22.3	−0.9	17.8	−21
B	mg/kg	28.8 *^a^*	31 *^a^*	8	40.8 *^b^*	42	40 *^b^*	39
Cu	mg/kg	6.8	10.3	51	11	62	8	18
Fe	mg/kg	59.5	76.5	29	85.3	43	68.3	15
Mn	mg/kg	45.3	48.3	6.7	53.5	18	42	−7.3
Zn	mg/kg	137.5 *^a^*	60.8 *^b^*	−56	52.8 *^b^*	−62	62.8 *^b^*	−54

Note: underlined numbers are significantly different. Different letters in rows indicate significant differences (*p* < 0.05).

**Table 5 plants-12-00372-t005:** Soil physical and chemical properties at a depth 0–30 cm (n = 16, July 2019).

Parameters	Units	Values	Parameters	Units	Values
Density	g/cm^3^	0.87 ± 0.02	Cr	mg/kg	78 ± 8
Porosity	% vol	66.0 ± 0.7	Cu	mg/kg	44 ± 6
Sand	wt%	34.6 ± 5.9	Fe	mg/kg	345 ± 4
Silt	wt%	13.3 ± 3.0	Hg	mg/kg	0.091 ± 0.005
Clay	wt%	52.1 ± 3.0	K	kg/ha	524 ± 43
Texture	-	Clay	Mg	kg/ha	1278 ± 188
Field capacity (FC) ^1^	v%	40 ± 3	Mn	mg/kg	22 ± 3
Wilting point (WP) ^1^	v%	29 ± 2	Na	mg/kg	895 ± 113
Total available water (TAW) ^1^	v%	12 ± 0.8	Ni	mg/kg	45 ± 5
TAW ^1^	mm	69.0 ± 3.5	Pb	mg/kg	7.0 ± 0.5
Organic matter	%	4 ± 2	P	kg/ha	312 ± 219
pH		7.8 ± 0.2	S	mg/kg	406 ± 152
Saturated hydraulic conductance	cm/h	0.12 ± 0.00	Se	mg/kg	<1.0
Ag	mg/kg	<2.0	SO_4_	%	0.008 ± 0.001
Al	mg/kg	1176 ± 129	Zn	mg/kg	85 ± 11
As	mg/kg	<5.0	Chloride	mg/kg	20 ± 5
B	mg/kg	0.5 ± 0.1	Sulphide	mg/kg	327 ± 159
Ba	mg/kg	193 ± 24	Organic carbon	%g/g	1.5 ± 0.4
Ca	kg/ha	5739 ± 622	Total carbon	%g/g	1.7 ± 0.6
Cd	mg/kg	<0.50	NH_4_-N	mg N/kg	8.0 ± 2.0
Co	mg/kg	16 ± 2	NO_3_-N	mg N/kg	1.8 ± 0.6

Note: ^1^: FC, WP, and TAW calculated according to Saxton and Rawls [53]; ± indicates standard deviation.

**Table 6 plants-12-00372-t006:** Physicochemical properties of aged leachate and groundwater used for the irrigation of willows in 2019 and 2020.

LEACHATE								GROUNDWATER			
Parameters	Units	n	Initial Characterization 2019	n	Average Aged Leachate Values 2019	n	Average Aged Leachate Values 2020	n	Average Groundwater Values 2019	n	AVERAGE Groundwater Values 2020
TSS	mg/L	3	57 ± 39	6	53 ± 37	8	79 ± 13	5	4.0 ± 2.4	8	1.8 ± 0.6
VSS	mg/L	3	23 ± 17	9	22 ± 12	8	31.0 ± 8.0	5	6.0 ± 2.2	8	<5
BOD5	mg/L	3	17. ± 8.2	6	31 ± 29	7	9.4 ± 3.0	5	5.2 ± 6.0	7	0.68 ± 0.67
COD	mg/L	3	204 ± 25	6	302 ± 17	9	288 ± 43	6	19 ± 29	9	9.8 ± 6.1
NH_3_-N	mg N/L	3	177 ± 32	6	260 ± 27	9	213 ± 35	6	0.17 ± 0.08	9	0.22 ± 0.06
NO_3_-N	mg N/L	3	0.09 ± 0.03	1	0.05	-	-	1	0.02	-	-
TKN	mg N/L	3	173 ± 46	1	240	1	180	1	1.1	1	1.1
o-PO_4_	mg P/L	3	0.270 ± 0.012	1	0.185	3	0.047 ± 0.073	1	0.0055	3	0.0050 ± 0.0003
P total	mg P/L	3	0.03 ± 0.01	1	0.020	3	0.103 ± 0.058	1	0.02	3	0.07 ± 0.04
Ca total	mg/L	3	202 ± 19	3	178 ± 23	2	232 ± 28	3	72 ± 8	2	79 ± 6
Cl	mg/L	3	198 ± 15	1	337	2	297 ± 85	1	36	2	40.0 ± 0.7
SO_4_	mg/L	3	20 ± 2	1	45	-	-	1	36	-	-
Al total	mg/L	4	0.033 ± 0.023	1	0.211	-	-	1	0.003	-	-
As total	mg/L	4	0.0012 ± 0.0005	-	-	-	-	-	-	-	-
Ba total	mg/L	4	0.370 ± 0.095	-	-	-	-	-	-	-	-
B total	mg/L	1	-	1	1.53	-	-	1	0.005	-	-
Cd total	mg/L	2	0.00012	-	-	-	-	-	-	-	-
Cu total	mg/L	4	0.001 ± 0.001	1	0.000025	-	-	1	0.000025	-	-
Fe total	mg/L	4	22 ± 14	4	24 ± 19	2	34.5 ± 5.0	3	0.63 ± 0.21	2	0.95 ± 0.50
Hg total	mg/L	4	<0.00001	-	-	-	-	-	-	-	-
K total	mg/L	3	147 ± 30	3	220 ± 36	2	268.0 ± 8.2	3	4.8 ± 0.5	2	4.5 ± 0.9
Mg total	mg/L	-	-	3	99 ± 10	2	122 ± 16	3	38.0 ± 5.5	2	22 ± 6
Mn total	mg/L	-	-	1	0.78	-	-	1	0.10	-	-
Na total	mg/L	4	170 ± 15	3	264 ± 32	2	333.0 ± 2.8	3	22 ± 2	2	22 ± 6
Ni total	mg/L	4	0.0075 ± 0.0030	1	0.013	-	-	1	0.000025	-	-
Pb total	mg/L	4	<0.003	-	-	-	-	-	-	-	-
Se total	mg/L	4	0.003 ± 0.002	-	-	-	-	-	-	-	-
Zn total	mg/L	3	0.34 ± 0.60	4	0.0015 ± 0.0017	2	0.006 ± 0.007	3	0.004 ± 0.006	2	0.0008 ± 0.0004
Phenols	mg/L	3	0.009 ± 0.005	2	0.009 ± 0.004	2	0.033 ± 0.01 0	2	0.001	2	<0.003
pH	-	-	-	5	6.8 ± 0.1	10	6.78 ± 0.21	5	7.3 ± 0.4	10	7.0 ± 0.2
EC	mS/cm	-	-	5	4.2 ± 0.4	10	3.76 ± 0.41	5	0.66 ± 0.10	10	0.62 ± 0.07
ORP	mV	-	-	4	−12.33 ± 78	10	39.4 ± 28.1	4	−29 ± 47	10	163 ± 89
TDS	mg/L	-	-	4	2077 ± 216	8	1893 ± 199	4	360 ± 98	10	307 ± 36

Note: BOD_5_ (biochemical oxygen demand); COD (chemical oxygen demand); EC (electroconductivity); ORP (oxidation-reduction potential); TDS (total dissolved solids); TSS (total suspended solids; VSS (volatile suspended solids); ± indicates standard deviation.

## Data Availability

Not applicable.

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
