# Peer review of "Treatment of Landfill Leachate by Short-Rotation Willow Coppice Plantations in a Large-Scale Experiment in Eastern Canada"

_plants, 2023, doi:10.3390/plants12020372_

Round 1
Reviewer 1 Report
GENERAL COMMENTS
The manuscript deals with the assessment of a plantation of willows for the treatment of landfill leachate. The authors have performed water, soil and biomass analysis to monitor the efficiency of the system.
SPECIFIC COMMENTS
Title
The word “treatment” appears twice in the title. Please, check this mistake.
1. Introduction
L. 40. Please, add a closing bracket after the word Canada.
2.1. Site description
L. 89. You mention that these plants were coppiced in fall 2018. Could you be more specific and provide the month/s instead? For instance, “coppiced in Xxxxx (September/October/November or December) 2018”.
2.2. Experimental setup and willow treatment
L. 105-108. The loadings applied are confusing. Here, in section 2.2, you state that D1 had a load of 3.7 mm/day and D2 had a load of 13 mm/d, whereas in the abstract you indicate that “D2 which was twice that of D1 (line 17)”, which implies that the load of D2 should be 2 x 3.7 = 7.4 mm/day. In addition, the treatment D2P, which is supposed to be similar to D2, presented a load of 7.5 mm/day (line 107), which is very different from the previously provided load of 13 mm/day for D2 (line 106). Please, check this paragraph.
L. 125-126. Why are the data from 2018 not presented?
2.4. Plant biomass
L. 154. The sentence “weighing three randomly selected fresh trees per plot and per treatment (60 trees, in total…)” is difficult to understand. According to Figure 1, there are 16 plots, and, if you pick three trees per plot, there would be 3 x 16 = 48 trees.
L. 155-156. There is a redundancy in the expression “a portable electronic electric scale”. Perhaps you meant “a portable electronic scale”.
2.5. Plant sampling and analysis
L. 166-172. Did you pick these leaves in 2019, in 2020 or in both years? Please, say it explicitly in the text.
2.7. Statistical analysis
L. 211. Which value of p was established to consider the existence of significant differences among samples?
3.4.4. Leaf nutrient analysis
L. 356-357. After talking about nitrogen, you state that “There was no clear statistical difference between treatments for the other elements”. However, in Table 6 one can see statistical differences for B and Zn too.
4. Discussion
L. 366. You have employed the exotic species Salix miyabeana. Why did you not employ a native species from Quebec? This might be more convenient from the environmental point of view.
L. 384-390. These sentences are difficult to understand. Please, check their structure.
L. 421. In the expression “1907 COD ha-1 yr-1”, the mass unit seems to be missing.
L. 472-473. Here, you indicate that “higher amounts of nutrients were measured in leaf tissues after leachate irrigation (especially B, Fe, N and K)”. However, in Table 6 it does not seem to be great variations in K content. Please, check it.
Table 3
Indicate the number of samples (n).
Why are there some values without standard deviation?
Table 4
Could you add standard deviations to efficacy values?
Figures 6 and 7
Please, indicate the parameters represented by the box-whiskers plot (average, median, standard deviation, etc.).
Author Response
Our answers are in italic
GENERAL COMMENTS
The manuscript deals with the assessment of a plantation of willows for the treatment of landfill leachate. The authors have performed water, soil and biomass analysis to monitor the efficiency of the system.
Thanks for this comment.
SPECIFIC COMMENTS
Title
The word “treatment” appears twice in the title. Please, check this mistake.
This has been corrected.
Introduction
- 40. Please, add a closing bracket after the word Canada.
This has been corrected.
2.1. Site description
- 89. You mention that these plants were coppiced in fall 2018. Could you be more specific and provide the month/s instead? For instance, “coppiced in Xxxxx (September/October/November or December) 2018”.
We have added that the plantation was coppiced in mid-November 2018.
2.2. Experimental setup and willow treatment
- 105-108. The loadings applied are confusing. Here, in section 2.2, you state that D1 had a load of 3.7 mm/day and D2 had a load of 13 mm/d, whereas in the abstract you indicate that “D2 which was twice that of D1 (line 17)”, which implies that the load of D2 should be 2 x 3.7 = 7.4 mm/day. In addition, the treatment D2P, which is supposed to be similar to D2, presented a load of 7.5 mm/day (line 107), which is very different from the previously provided load of 13 mm/day for D2 (line 106). Please, check this paragraph.
- 125-126. Why are the data from 2018 not presented?
The beginning of paragraph of the section 2,2 has been rewritten to specify the information. We think that it is clearer as it is now mentioned.
The 0.4 ha strip-block experimental setup was divided into four blocks (B1-B4) comprising a total of 20 equally sized 9.14 m x 22 m rectangular plots, with four treatments (D0, D1, D2 and D2P) replicated four times (Figure 1). Each plot was assigned one of four treatments corresponding to a maximum irrigation rate of 3 m3 day-1 of leachate for D1, 6 m3 day-1 of leachate for D2, 6 m3 day-1 for of leachate plus 40 kg P2O5 ha-1 yr-1 of phosphorus fertilizer (D2P) or 3 m3 day-1 of water for D0. Plot assignments for treatments D1 and D0 were selected based on the irrigation lines system design, while plot assignments between treatments D2 and D2P were randomized (Figure 1).
2.4. Plant biomass
- 154. The sentence “weighing three randomly selected fresh trees per plot and per treatment (60 trees, in total…)” is difficult to understand. According to Figure 1, there are 16 plots, and, if you pick three trees per plot, there would be 3 x 16 = 48 trees.
The reviewer is right. This was our mistake, we had 16 plots which makes 48 samples trees in total.
- 155-156. There is a redundancy in the expression “a portable electronic electric scale”. Perhaps you meant “a portable electronic scale”.
This has been corrected.
2.5. Plant sampling and analysis
- 166-172. Did you pick these leaves in 2019, in 2020 or in both years? Please, say it explicitly in the text.
This has been corrected and we specify that it has been done in 2020.
2.7. Statistical analysis
- 211. Which value of p was established to consider the existence of significant differences among samples?
Tukey post-hoc tests were performed when the ANOVAs detected significant differences at the traditional threshold of 5%. This information has been added in the statistical analysis paragraph.
3.4.4. Leaf nutrient analysis
- 356-357. After talking about nitrogen, you state that “There was no clear statistical difference between treatments for the other elements”. However, in Table 6 one can see statistical differences for B and Zn too.
This is right, the correction has been made.
- Discussion
- 366. You have employed the exotic species Salix miyabeana. Why did you not employ a native species from Quebec? This might be more convenient from the environmental point of view.
We have experimented several trials involving native willows and they revealed to be generally less productive than the selected exotic willow cultivars like the ones used in this study. The production of biomass is an essential criterion for optimizing plant evapotranspiration and pollutant removal capabilities in wastewater. The more plants produce biomass, the more they perform for these processes. Cultivars used in this study do not pose any environmental issue in this specific context.
- 384-390. These sentences are difficult to understand. Please, check their structure.
Changes have been made to clarify these sentences.
- 421. In the expression “1907 COD ha-1 yr-1”, the mass unit seems to be missing.
It has been corrected.
- 472-473. Here, you indicate that “higher amounts of nutrients were measured in leaf tissues after leachate irrigation (especially B, Fe, N and K)”. However, in Table 6 it does not seem to be great variations in K content. Please, check it.
The reviewer is right, K has been removed in the sentence.
Table 3
Indicate the number of samples (n).
Why are there some values without standard deviation?
Table 3 has been redone to include number of samples.
In each experimental block (4 in total in the experimental design), there was one plot per treatment. For Al, B, Ca, Cu, Fe, K, Mg, Mn, P and Zn soil content determination, four soil samples were collected per plot and homogenized in only one analytical sample, giving finally four analytical samples per treatment (one per block) in June and in October 2020 (n = 4). For As, Ba, Cd, Co, Cr, Ni, Pb and Se, the same sampling approach was made except that three soil samples were collected per plot and per treatment and homogenized in only one analytical sample, giving finally one analytical sample per treatment for the whole experimental design (n=1).
Table 4
Could you add standard deviations to efficacy values?
In total, 9 porewater sampling campaigns were made in 2020 at 15 day intervals (from June 04 to October 08). One lysimeter was installed in each experimental plot for a total of 4 lysimeters per treatment (D0, D1, D2 and D2P). So, in theory, we should get a total of 36 values for each treatment during the season. But, during sampling campaigns, we found that 1, 2 or 3 lysimeters per treatment were empty probably due to the high absorption rate of willows. So, we decided to gather data of each treatment in order to calculate a seasonal average mean and standard deviation from which we calculated a seasonal percentage of efficacy. This represents a total of 26 values for D0 and 28 values for D1, D2 and D2P. It is for that reason we did not present standard deviations to efficacy values.
Figures 6 and 7
Please, indicate the parameters represented by the box-whiskers plot (average, median, standard deviation, etc.).
Corrected.
Reviewer 2 Report
In the present manuscript phytofiltration of mature landfill leachate was examined. The introduction contains general information on landfill leachate, however does not focus on the characteristics of mature leachate and there is no reference on other non conventional treatment systems. Even-though the BOD/COD ratio is low the pH of the influent is acidic and EC is low despite the presence of sodium cations and ammonium content. Could these findings explained by the authors? The design of the research is appropriate and the findings are supported. The manuscript lacks molecular analysis and the authors speculate regarding the presence of the microbial community.
Author Response
Our answers are in italic.
I do find this work interesting and valuable. The manuscript is well written. Some small suggestions:
Line 39 – the bracket is missing after “Canada”
This has been corrected.
Tables 1 and 2, 3, 4, 5 – what these values after ± mean? (Standard error?) - it should be explained
These comments, mentioned by reviewer 1, have been considered and corrections have been made.
Line 178 – “morethan” should be written separately
It has been corrected.
Figures 5, 6 and 7 – letters for significant differences should be better connected with groupings. Moreover, it should be explained what these bright and dark circles mean as well as whiskers, horizontal lines and rectangles. What kind of statistical values they represent?
Letters were moved so to be better connected with groupings. Box plots are commonly used to illustrate results with some variability. Classically, each box is made up of a minimum value, three quartiles (the second of which is the median - horizontal bar-) and a maximum value. Isolated circles are outliers/extreme values.
For figure 5B – I do not see values for D0 – October
This value was very low (0.01). So, it was at the level of X axis.
Line 356 – That is not true – according the Table 6 there were differences for B and Zn.
Corrected. We have modified the text.
My suggestion is also to underline better what is new in your research.
A sentence has been added in the conclusion to emphasize the aspect of novelty.
Reviewer 3 Report
Dear authors,
I do find this work interesting and valuable. The manuscript is well written. Some small suggestions:
Line 39 – the bracket is missing after “Canada”
Tables 1 and 2, 3, 4, 5 – what these values after ± mean? (Standard error?) - it should be explained
Line 178 – “morethan” should be written separatelly
Figures 5, 6 and 7 – letters for significant differences should be better connected with groupings. Moreover, it should be explained what these bright and dark circles mean as well as whiskers, horizontal lines and rectangles. What kind of statistical values they represent?
For figure 5B – I do not see values for D0 – October
Line 356 – That is not true – according the Table 6 there were differences for B and Zn.
My suggestion is also to underline better what is new in your research.
Author Response
Our answers are in italic.
In the present manuscript phytofiltration of mature landfill leachate was examined. The introduction contains general information on landfill leachate, however does not focus on the characteristics of mature leachate and there is no reference on other non conventional treatment systems.
We have added a comment in the introduction about this.
Even-though the BOD/COD ratio is low, the pH of the influent is acidic and EC is low despite the presence of sodium cations and ammonium content. Could these findings explained by the authors?
In the present study, the ratio BOD/COD was around 0.03, indicating a low biodegradability, a typical characteristic of stabilized leachates (> 10 years; BOD/COD <0.1 and COD <0.5 g L-1) of inoperative landfill cells. We agree that this type of leachates has typically a pH > 7.5. The nature of waste material as well as climatic conditions and operating practices of the landfill site may influence the final composition of leachates and therefore the pH measured in the leachate. Regarding the EC: the dissolved Na ions were also measured (data not shown). This fraction represented about 65% of the represented a part of the total one. This may explain why EC data were lower than expected. Further, there was a good correlation between EC and TDS, in particular for sodium.
The design of the research is appropriate and the findings are supported.
The manuscript lacks molecular analysis and the authors speculate regarding the presence of the microbial community.
We agree but this was not really part of the study.